# Local features drive identity responses in macaque anterior face patches

Elena N. Waidmann[1,3,6], Kenji W. Koyano [1,6] ✉, Julie J. Hong[1,4], Brian E. Russ [1,5] & David A. Leopold [1,2] ✉

Humans and other primates recognize one another in part based on unique structural details of the face, including both local features and their spatial configuration within the head and body. Visual analysis of the face is supported by specialized regions of the primate cerebral cortex, which in macaques are commonly known as face patches. Here we ask whether the responses of neurons in anterior face patches, thought to encode face identity, are more strongly driven by local or holistic facial structure. We created stimuli consisting of recombinant photorealistic images of macaques, where we interchanged the eyes, mouth, head, and body between individuals. Unexpectedly, neurons in the anterior medial (AM) and anterior fundus (AF) face patches were predominantly tuned to local facial features, with minimal neural selectivity for feature combinations. These findings indicate that the high-level structural encoding of face identity rests upon populations of neurons specialized for local features.

Humans are adept at recognizing individuals, as well as reading their moods, gestures, and intentions, based on a visual analysis of the face. Face perception is greatly enhanced in primates compared to other mammals, owing in part to the rise of vision as the dominant social sense[1]. The primate temporal lobe contains specialized regions of cortex engaged in the structural analysis and perception of faces[2–6], including the recognition of individual identity[7–11]. The coding principles underlying this analysis and the interaction of these areas with other networks involved in social behavior are active topics of research in cognitive neuroscience.

Primate faces share a common first order configuration, including paired eyes centered above a nose and mouth in the context of the head and body. Psychophysical studies of human face recognition often contrast the relative importance of local feature details, such as the eyes or mouth, vs. the configuration of these elements relative to one another within the whole face and head[12,13]. While experimental evidence has demonstrated that both facets are used to determine identity[14–16], most research has placed emphasis on configural processing, in part because it appears more important for the recognition of faces than for other objects. For example, swapping or even slightly displacing internal facial features can severely disrupt the recognition of identity[17–19]. Likewise, inversion disproportionately impacts the recognition of faces, which is thought to reflect a reversion to feature-based analysis[20–22]. The importance of configural processing is also reinforced by psychophysical studies carried out using the framework of an abstract face space, where changes in identity accompany continuously varying geometrical distortions[23–25].

Single-unit studies from the macaque face network are mixed in their support of holistic vs. parts-based analyses of faces. Some neurons appear distinctly sensitive to face parts. For example, many cells in the posterolateral (PL) face patch, which is considered to be the entry point to the macaque face network[26], are specifically tuned to the contralateral eye[27]. In the middle face patches, thought to represent a higher stage of processing, neurons exhibit combined tuning profiles

[1]Section on Cognitive Neurophysiology and Imaging, National Institute of Mental Health, 49 Convent Dr., Bethesda, MD 20892, USA. [2]Neurophysiology Imaging Facility, National Institute of Mental Health, National Institute of Neurological Disorders and Stroke, National Eye Institute, 49 Convent Dr., Bethesda, MD 20892, USA. [3]Present address: Laboratory of Neurogenetics of Language, The Rockefeller University, 1230 York Avenue, New York, NY 10065, USA. [4]Present address: Milton S. Hershey Medical Center, Penn State College of Medicine, 700 HMC Crescent Road, Hershey, PA 17033, USA. [5]Present address: Center for Biomedical Imaging and Neuromodulation, Nathan S. Kline Institute for Psychiatric Research, 140 Old Orangeburg Road, Orangeburg, NY 10962, USA. [6]These authors contributed equally: Elena N. Waidmann, Kenji W. Koyano. ✉e-mail: kenji.koyano@nih.gov; leopoldd@mail.nih.gov

that include the details of local facial features as well as their geometrical arrangements[8]. The responses of neurons in these regions are contingent on face context and are sensitive to face inversion[8,28], suggesting a contribution of holistic processing. At a yet higher level, neurons in the anterior medial (AM) face patch have been shown to encode face identity in a more invariant manner, responding selectively to images of individuals taken from different viewpoints[9]. In area AM and in the anterior fundus (AF) face patch, other experiments using continually morphed photorealistic faces emphasized the role of continuous geometrical distortion in the tuning for face identity, which may be more closely linked to the concept of configural analysis than parts-based analysis[7,10,11]. Together, these findings have suggested that, within the face patch network, the early stage analysis of faces in PL is focused on the details of certain individual features, but, following a transition in the middle face areas, the later stage analyses in AM and AF operate on holistic and configural aspects of the face. However, in those studies, the relative contributions of holistic and parts-based face processing were not systematically examined. Might the identity tuning of neurons in anterior face patches be driven, in part, by local facial features?

Here we examine the encoding of individual local features as well as their combination within the entire face, head, and body among neurons in macaque AM and AF face patches. We present a novel method in which facial and image features are excised and systematically recombined into a new photorealistic image. This recombination allows us to evaluate the neural tuning to individual facial features within multiple identity contexts. We report an unexpected sensitivity of face-selective AM and AF neurons to single facial components, with minimal evidence for tuning to conjoined features at the single neuron level. Control experiments further show that neurons persisted in their responses to a favored local facial feature that was physically transplanted into a new face, body, and scene, even if that feature comprised as little as 0.3% of the image stimulus. We discuss how this apparent parts-based encoding within the face network might be reconciled with broad evidence supporting the importance of holistic face processing for identity recognition.

## Results

We examined the contribution of facial and body parts to neuronal responses by systematically swapping select parts of photorealistic images of conspecifics (Fig. 1). Ten high-resolution photographs of monkeys (Fig. 1b) were split into four component parts (Fig. 1a) and then recombined into four categories of pairwise feature recombination: eyes × mouth, inner × outer face, head × body and monkey × scene (Fig. 1c, d). These swapping categories resulted in 100 novel inner faces, heads, monkeys, and whole scenes, respectively (Fig. 1d). The term 'eyes' was used to label the upper region of the inner face, including the eyes, portions of the surrounding skin, eyebrows, and forehead. The term 'mouth' pertained to the lower inner face, including the lips, jaw, and nose (Fig. 1a). While it is possible to dissect these regions further, the current study used upper and lower inner face components in its combinatorial parts analysis. Importantly, each recombined image appeared natural and realistic, including the hybrid monkey faces. We presented both recombined and part-alone images on the screen at three different sizes (Fig. 1e). In total, the stimulus set consisted of 1350 images (see Fig. S1a for exemplars), each of which was presented at least 10 times (13,500 individual trials) for each of the cells included in this study. To achieve the exposure of single units to this large number of trials, we used microwire brush arrays specialized to stably hold neurons across sessions[29]. The brush arrays were implanted chronically into fMRI-defined AM and AF face patches (Fig. 1f). Stimulus presentation was carried out over the course of 1–2 weeks as the neurons were monitored. The same neurons were identified across multiple days by assessing the waveform similarity and the selectivity of responses for a familiar stimulus set, that did not

overlap with the test stimulus set (Fig. S2). We recorded from 403 neurons meeting this criterion, of which 208 neurons (AM: 80 neurons, AF: 128 neurons) were face-selective ($p < 0.05$, one-way ANOVA).

### Tuning of anterior face patch neurons is dominated by local features

Neurons in both AM and AF were strongly tuned to specific local facial features, such as the eyes of one or two individual monkeys. For example, Fig. 2a, b shows the responses of a neuron in AM whose activity was driven strongly by the eye component of the face and was primarily responsive to the eyes derived from monkey m9. Figure 2a shows that the eye image extracted from this monkey stimulus led to stronger responses relative to those of monkey m10, regardless of the pairing of these eyes with other mouths, or of their context within other heads, monkey bodies, or scenes. The pattern of responses for all pairs of eyes and corresponding stimulus combinations is summarized for this neuron in Fig. 2b, which underscores the dominance of select pairs of eyes in stimulating the neuron (two-way ANOVA; portion of explained variance due to eyes = 0.38, mouth = 0.02, interaction = 0.02).

Similar dominance of single image features, including both inner and outer facial features, was common across both the AM and AF neural populations using this recombination approach. For example, Fig. 2c shows an AF neuron responding only to images that contained the mouth from monkey m4, with no other image part offering strong response modulation (two-way ANOVA; explained variance due to eyes = 0.01, mouth = 0.47, interaction = 0.01). Figure 2d shows an AF neuron that showed minimal response to internal faces but was strongly driven by two outer face exemplars (outer faces 1 and 9; two-way ANOVA; explained variance due to inner face = 0.03, outer face = 0.34, interaction = 0.04). As in the first example, the critical features for these two neurons dictated their responses even when those features were embedded within a wide range of different contexts. Examples of other eyes-, mouth-, and outer-face-selective neurons can be found in Fig. S3.

We next evaluated the importance of critical image features across the face-selective neural population. The hybridization approach allowed us to systematically quantify for each neuron the proportion of response variance explained by each face or image part. For each category of stimulus combination (e.g., eyes vs. mouth), we performed a 2-way ANOVA to assess the contribution to response variance by each of the two combined image components as well as their conjunction (i.e., the interaction term). As an example, the eyes-selective neuron shown in Fig. 2a, b owed 38.5% of its response variance to changes to the eyes, 1.9% to changes to the mouth, and 0.2% to the interaction of the two features.

Across the population, the responses of most neurons were dominated by a single facial feature, with the level of dominance and specific facial feature varying across the different recording sites. This is evident when assessing their response variance explained by different components of the face (Fig. 3). Figure 3a shows this measure for each of the recorded neurons under each of the four main image recombination conditions. For neurons in three of the recording locations (top three rows), responses were dominated by components of the inner face, namely the mouth or eyes, where for two of the AF sites (bottom two rows), there were stronger contributions from the outer face and body. Importantly, very little variance was accounted for by the interaction between image components, highlighted in Fig. 3b.

The focus of neural selectivity upon individual features, rather than their combination, was particularly evident when considering the variance explained by the eyes, mouth, and outer face. Upon separating subpopulations that showed any explained variance preference for each of these parts (135/208 of the face-selective neurons, see Methods), cells demonstrated highly selective tuning to these single local features (Fig. 3c). The eyes- and mouth-selective cells both

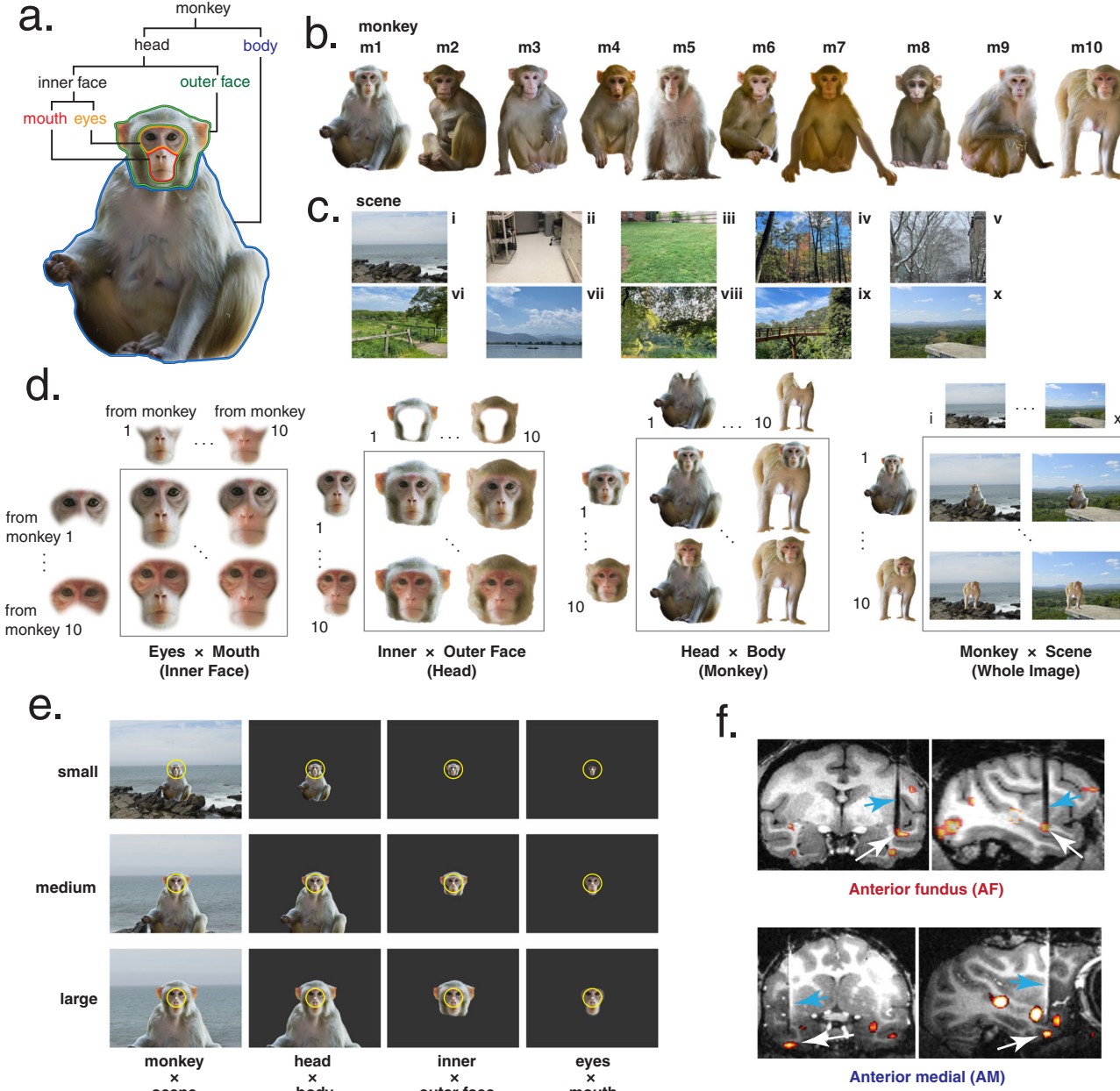

**Fig. 1 | Recombined parts stimuli. a** Photograph of one macaque monkey broken down into component parts. Eyes and mouth compose the inner face, outer and inner face compose the head, and body and head compose the full monkey. Macaque images were obtained from https://figshare.com/articles/dataset/Macaque_Faces/9862586/1 under a CC-BY 4.0 license. **b** Example high-resolution photos of front-facing macaques which form the basis of the stimulus set. **c** Example high-resolution background scene photographs (taken by authors). **d** Parts from monkeys 1 and 10 systematically recombined into the four swapping categories: eyes × mouth, inner × outer face, head × body, and monkey × scene. **e** The three stimulus presentation sizes for each swapping category, and central fixation point (white dot) with permissible fixation window (yellow line; not visible to the monkey). Macaque and scene images in **a**–**e** are not original stimuli, but are mock stimulus images representative of the original stimulus set. **f** Functional overlay of face patches AF and AM from two monkeys, with tracts of chronic recording electrodes (blue arrow) and the targeted patches (white arrow) indicated.

displayed minimal tuning for the opposite internal facial feature, and the outer face-selective cells showed low tuning to the internal face. The variance explained by eyes, mouth, and outer face for every face-selective cell was further plotted for visualization purposes in Fig. 3d, and the responses of most cells followed one of the three axes, showing little conjoint selectivity.

These three populations (see Fig. S4a, b for the parts-selective and non-parts-selective units) were not strictly divided between the two face patches. While eyes-selective units predominated at both AM recording sites (22/44 MA, 24/36 WA), outer face- and mouth-selective cells were also present in smaller proportions (Table S1). Face patch AF

was more heterogeneous in its neural tuning profiles and differed somewhat across AF recording locations, with mouth-selective cells predominating at one site (19/36 MO) and outer face cells at the others (19/37 SP1, 25/55 SP2) and other selectivity types minimally represented (Table S1). However, across all sites the single local features dominated the responses to faces across dramatic changes in image context, with little evidence for combinatorial or holistic tuning.

**Responses to local features predict responses to whole images**
Given the extent to which individual local features shaped the responses to recombined images, we next analyzed the extent to

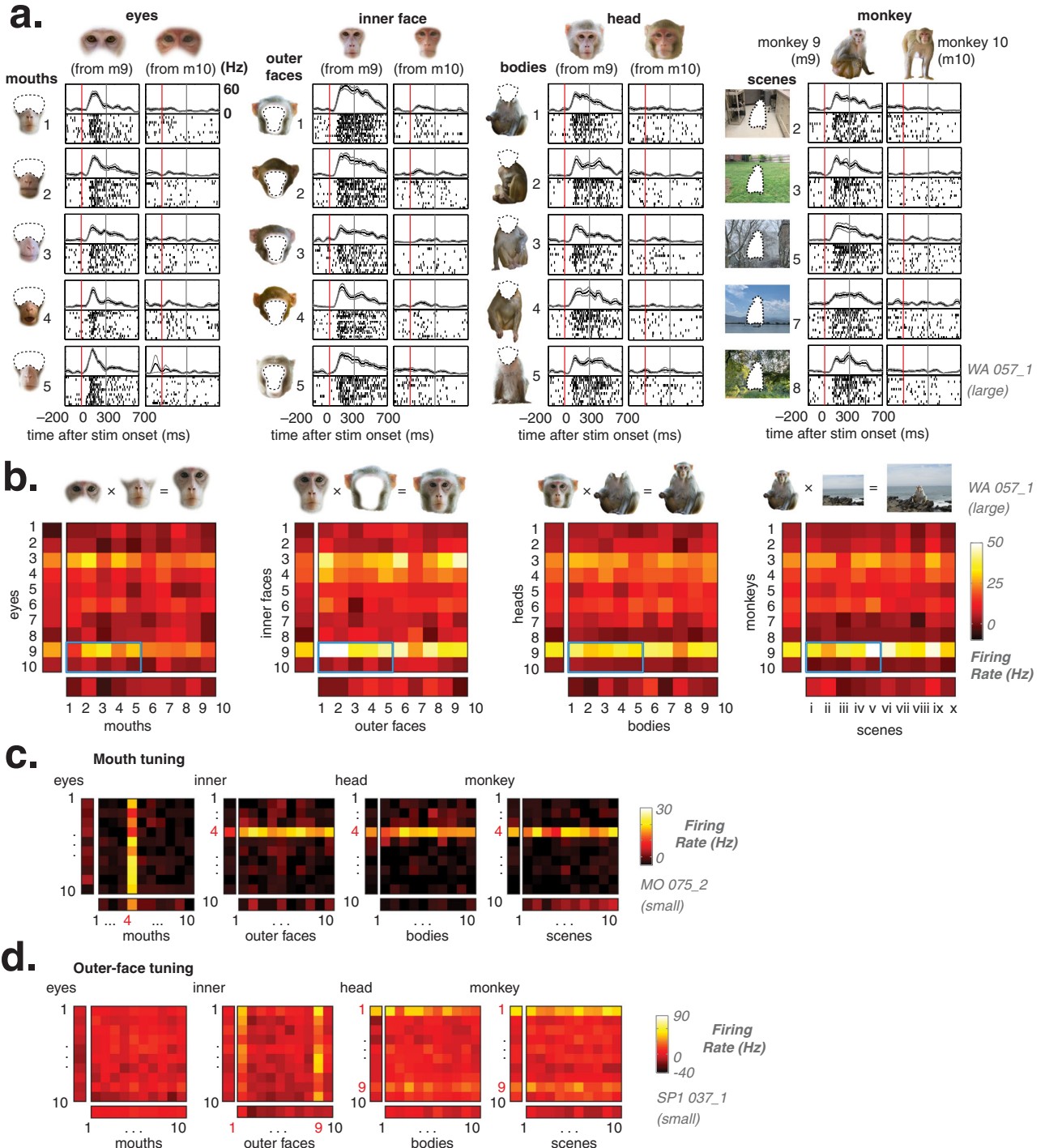

**Fig. 2 | Individual face-selective neurons show strong tuning to single local facial parts. a** Raster plots showing response of one typical AM neuron (WA-057_1). From left column: response to *eyes × mouth* stimuli (eyes from monkeys 9 and 10, recombined with mouths from monkeys 1 to 5). Subsequent columns: responses to *inner × outer face*, *head × body*, and *monkey × scene* stimuli from monkeys 9 to 10. **b** Heatmaps of AM neuron WA-057_1 average firing rate in response to large stimuli of all swapping categories; responses to combined stimuli in central square

heatmaps, parts alone stimuli in flanking rectangle heatmaps. Responses in **a** outlined in blue. Tuning to eyes persists across swapping categories. Macaque images in **a**–**b** were obtained from https://figshare.com/articles/dataset/Macaque_Faces/9862586/1 under a CC-BY 4.0 license. **c** Heatmaps of AF cell MO-074_1 responses to all small stimuli. Persistent tuning for stimuli that contain mouth 4. **d** Heatmap of AF cell SP-037_1 responses to all small stimuli indicating a strong preference for outer faces 1 and 9.

which the neural responses to local face features, including those displayed in isolation, would predict responses to highly complex images containing those features. To address this question, we applied three different approaches.

In the first approach, we compared neural responses to local features to those obtained within wider image contexts. Figure 4a

outlines this method applied to neurons in the eyes-selective population (see Fig. 3c, Methods for population definitions). For each neuron, the mean tuning for each set of eyes (while embedded within an internal face) was computed, and the responses of the population were ordered according to the strongest eye identity preferences of each cell (Fig. 4a). This cell order was then applied to the mean feature

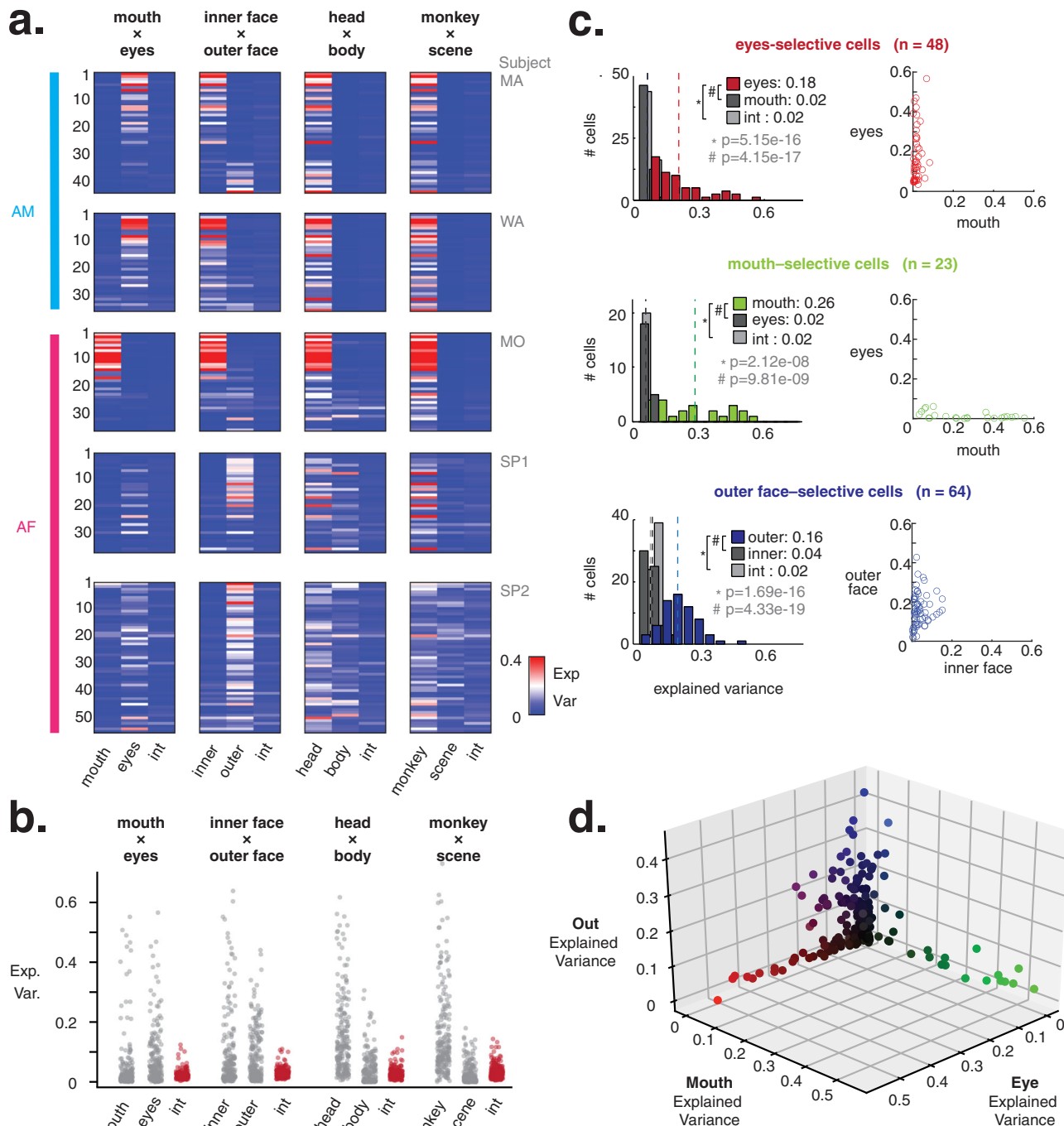

**Fig. 3 | AM and AF neurons show strong parts-based tuning. a** Heatmaps of response variance (2-way ANOVA) explained by each swapping factor and interaction (e.g., eyes, mouth, eyes × mouth) for face-selective cells at AM (blue) and AF (pink) sites. Most cells show high explained variance by one feature per swapping group. **b** Portion of response variance explained in each swapping category by the respective swapping factors (gray) and by the interaction term (red). Variance explained by the interaction of parts is low across all sites and all swapping categories. **c** Classification of neurons as eyes-selective (top, red), mouth-selective (middle, green), or outer face-selective (bottom, blue). Left, histograms of variance explained by primary features (eyes, mouth, outer face), secondary features (mouth, eyes, inner face), and the interactions. Text insets: mean explained variance by each factor; means also marked with dotted line. The results indicate significantly more tuning to each primary feature (Wilcoxon Signed Rank test, *p* values listed in gray). Right, scatter plots of explained variance due to primary and secondary features for each cell. **d** Response variance explained by eyes, mouth, and outer face, for all 208 face-selective cells. Neurons primarily fall along one of the three axes, indicating that most are tuned to a single local feature.

responses across subsequent swapping categories (e.g., mean responses for each monkey, embedded within a scene, Fig. 4a). Figure 4b shows this approach applied to the entire eyes-selective population, across the different recombination categories and image parts. Neurons in each panel are ordered from left to right based solely

on cells' preferences for the different eyes identities. The similar pattern across the four top panels indicates that the mean eyes response was an excellent predictor of neural responses across different image contexts, even the full *monkey × scene* images. This similarity can be seen further in the correlation matrix in Fig. 4c, which shows that the

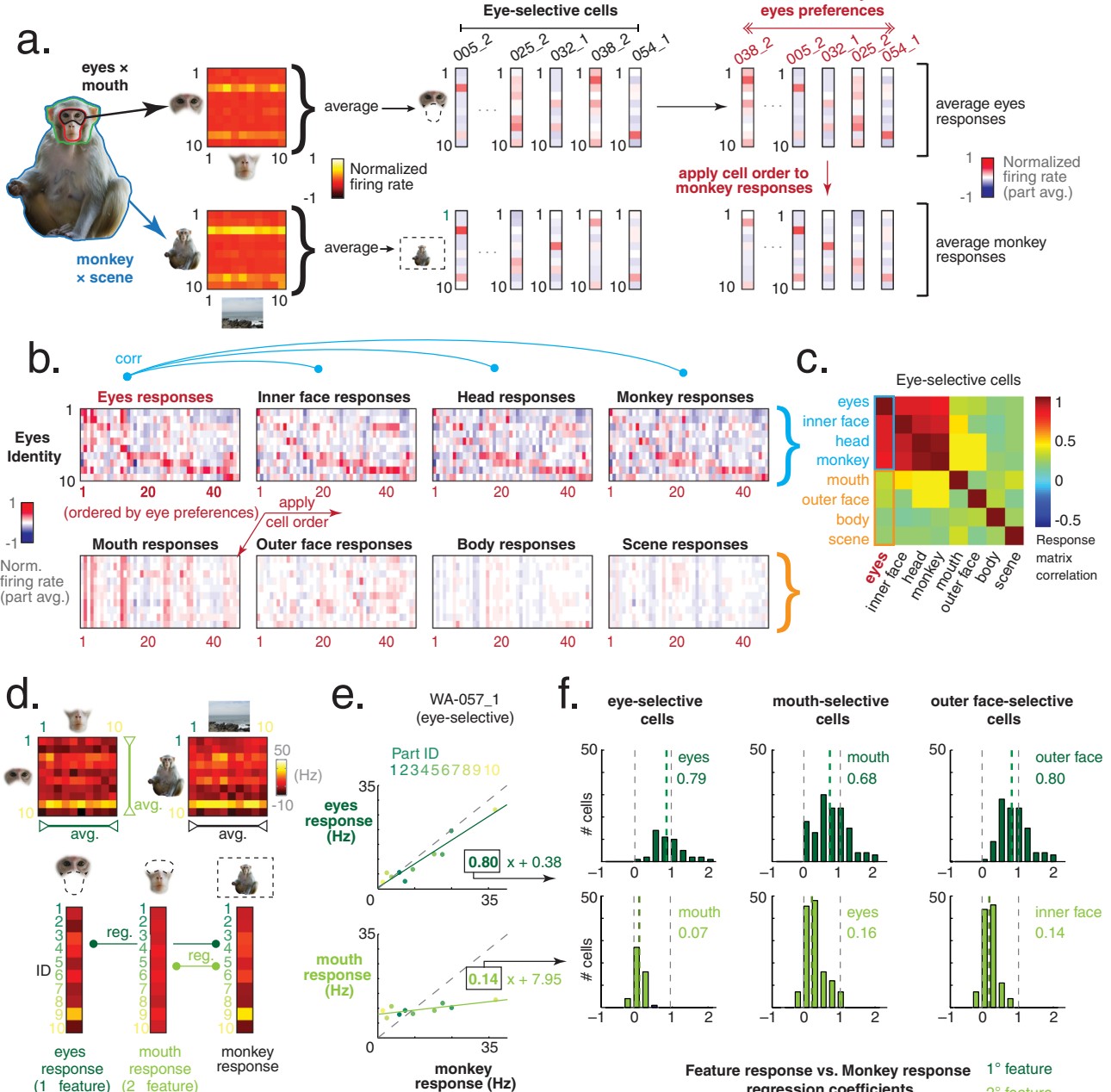

**Fig. 4 | The responses to local parts predict responses to full naturalistic images. a** Eyes-selective cells: From responses to eye × mouth and monkey × scene stimuli, average eyes and average monkey responses. Population ordered by the cells' eye identity preferences; cell order subsequently applied to the monkey responses. **b** Eyes-selective cells: Heatmaps of average firing rates for each local part. As in **a**, cells in each response matrix are ordered by eyes identity preferences. **c** Eyes-selective cells: Pairwise correlations between part response matrices. Correlations between eyes responses and sorted inner face, head, and monkey response matrices are high (top row in **b**, outlined in blue). Correlations between eyes response matrix and sorted mouth, outer face, body, and scene responses are lower (bottom row in **b**, outlined in orange). **d** Eyes-selective cell WA-057_1: Average responses to eyes and mouths; compared to average response for monkeys.

Macaque images in **a** and **d** were obtained from https://figshare.com/articles/dataset/Macaque_Faces/9862586/1 *under a CC-BY 4.0 license*. **e** Eyes and mouth response vectors from **d** plotted against the response vector for monkeys. Colors of points indicate specific identity (1–10); dashed gray line shows unit line. Eyes response predicts monkey response (dark green, top, linear regression, $r^2 = 0.92$) better than mouth response (light green, bottom, linear regression, $r^2 = 0.55$). **f** Linear regression between monkey responses and primary (dark green: eyes, mouth, outer face) and secondary (light green: mouth, eyes, inner face) local part responses. Distribution of regression coefficients (boxed in **e**) for eyes-selective, mouth-selective, and outer-face-selective neurons. Dashed lines show median regression coefficients (green), and coefficients of 1 and 0 (gray).

tuning for eyes is highly correlated with the neural tuning across all larger image contexts. In contrast, the bottom right quadrant of Fig. 4c indicates that facial components that do not contain the eyes had far less contribution to neural tuning across higher image contexts. Similar analyses applied to the populations of mouth-selective and outer-face-selective cells also show that these critical local image parts

drove neural responses all the way up to the level of full *monkey × scene* images, shown in Fig. S4g.

In the second approach, we performed a linear regression analysis to quantify the contribution of local features to the neural tuning for the compete monkey identity image (Fig. 4d–f). In the eye-selective example neuron shown in Fig. 4d, e, the average

responses to each pair of eyes (embedded within an inner face) closely resembled the responses to the full monkeys containing those eyes ($r^2 = 0.92$). By contrast, for the same group of eyes-selective neurons, the average responses to each mouth were much less predictive of the corresponding responses to the full monkeys, though still positively correlated ($r^2 = 0.55$). For each cell, we performed a linear regression comparing the responses to full monkeys to the average responses to primary features (eyes, mouth, outer face) for each tuning type (eyes-selective, etc.), to ask how well a response to one set of eyes embedded in just an inner face, for example, would predict the response to a full monkey image with those eyes. Across the population of eyes-, mouth-, and outer face-selective neurons, the average regression coefficients comparing the primary features against monkey responses were near 1.0, indicating that the responses of neurons to full monkey identity were not only qualitatively predicted by local features but also quantitatively predicted (Fig. 4f). The regression coefficients for the primary features were in all cases significantly higher than that of their counterpart feature (Wilcoxon Sign Rank test, $p < 0.05$). We also directly compared responses to isolated and combined parts stimuli for all parts-selective cells (Fig. S5a). There was a range of preference for isolated or combined stimuli, with most of the parts stimuli represented roughly similarly across both conditions (Fig. S5b). Eyes-selective cells as a whole showed little preference between isolated and combined stimuli. Mouth-selective and outer face-selective cells showed a slight but significant (t-test, $p < 0.01$) preference for features in combination for some swapping groups (Fig. S5b), while still owing their identity tuning primarily to the mouth or outer face.

In the third approach, applied to a population of predominantly eyes-selective AM neurons in one monkey, we measured the responses to the ten isolated eyes on a blank screen and then transplanted them into an entirely different monkey occupying a background scene (Fig. 5a). The stimuli were presented at three different spatial scales (Fig. S1c, d). As in the pairwise recombination experiments, neurons in this area exhibited persistent tuning to individual eye exemplars. This tuning was preserved when the eyes were transplanted into entirely different identities and scenes (Fig. 5b, c). When we compared neural responses to the isolated eyes to those in which the eyes were embedded within the full image context, we found that the tuning pattern to the eyes alone strongly predicted the response pattern to the full image (Figs. 5e, S6). To quantify this effect, we performed a two-way ANOVA (as in Fig. 3a) and, as in the recombination experiments, found a large proportion of cells with higher response variance for the eyes, with little contribution from larger image context or the interaction (Fig. 5f, g). Moreover, this preserved tuning persisted amid a threefold change in the linear scaling of the image in the tuning of individual neurons (Fig. 5f).

Together, these experiments and analyses demonstrate that the individual local face components can strongly predict the responses of neurons in anterior face patches to rich, complex images containing full monkeys and background scenes. In the case of the smallest presentation, the eye component of the face that dominated the tuning of the neuron's response amounted to only 0.3% of the pixels displayed in each image (Fig. 5).

## Discussion

The results from this study indicate that the identity-tuning of neurons in anterior face patches AF and AM in the macaque is dominated by local facial features. The pairwise recombination and single feature transplantation converged on the finding that neural responses in these areas were so reliably dictated by local face components that the presentation of isolated feature exemplars often provided a strong quantitative prediction of neural responses to much more complicated stimuli, including for synthetic stimuli in which local feature were transplanted into entirely different identities.

### Unexpectedness of results

These findings come as a significant surprise. For one, the complex feature selectivity within the cortical visual hierarchy is generally conceived to progress from local, at more posterior locations, to holistic, at more anterior locations[30,31]. This trend is believed to follow the progression of receptive field size, feature complexity, and image invariance. Within the face patch system, this broad conception is supported by findings such as the dominance of the contralateral eye as a local feature component in posterior region PL[27] and the view invariance of face identity tuning in anterior region AM[9]. This progression also seems consistent with combined effects of local features and face context within the middle face patches[8]. While other findings, such as the well-behaved neural tuning for morphed face identity in the context of face space[7,10,11], do not directly address the role of individual facial features, they seem to favor more holistic modes of identity processing.

These previous physiological observations, together with psychophysical evidence indicating the importance of holistic and configural processing for faces[13], predicted that neurons in anterior face patches, at least face patch AM, would have been most strongly driven by conjunctions or configurations of facial features. We set out to investigate this prediction, but found that AM and AF neurons had near negligible encoding of feature combinations, with their responses instead dominated by local feature details.

The most common tuning profile in patch AM involved the upper face, with more varied tuning profiles within AF (Fig. 3a, Table S1). Heterogeneous selectivity for different facial parts was also previously reported in ML[8] and AM[32] face patches. The AF tuning was more heterogeneous and also varied somewhat across recording sites, which may reflect different functional subdivisions within the face patch, or differences in the specialization across animals. The high proportion of the outer face-selective neurons in AF was unexpected, in part because the fundus of the STS is thought to be specifically adapted for configurable facial features that confer dynamic social information, most notably the eyes and mouth[33–36]. The outer face represents a fixed and immutable cue for face identity, and, aside from the ears and some postural cues, has only a secondary role in social signaling[37]. While the specificity of outer face tuning was slightly weaker than for eyes and mouth (see Fig. 3c, d), the responses of many AF neurons were dominated by outer face information in the recombination and transplantation paradigms. Since the external features of faces are more important for face processing in young children[38,39] and configural processing for the internal face develops slower than feature-based face processing[40], the parts-based tuning observed in AM and AF face patches may reflect a perceptual mechanism that develops earlier in the brain. Future investigations should consider the outer face as a significant contributor to neural tuning in this region. In sum, the AM and AF face patches showed potentially important differences in the specific features to which they were tuned, but crucially, neither appeared to be an integration site for holistic face information.

### Potential explanations and relationship to previous psychophysical studies

Clues for how to reconcile the present results with our current understanding of object processing may come from studies emphasizing non-holistic aspects of face processing in the anterior temporal cortex. For example, evidence from human fMRI studies suggests that significant holistic processing occurs in the posteriorly located occipital and fusiform face areas[16,41,42]. However, recent evidence using composite faces suggest that more anterior temporal face areas, thought to correspond to macaque anterior face patches[43,44], may be less sensitive to holistic information[45]. Thus, in agreement with our

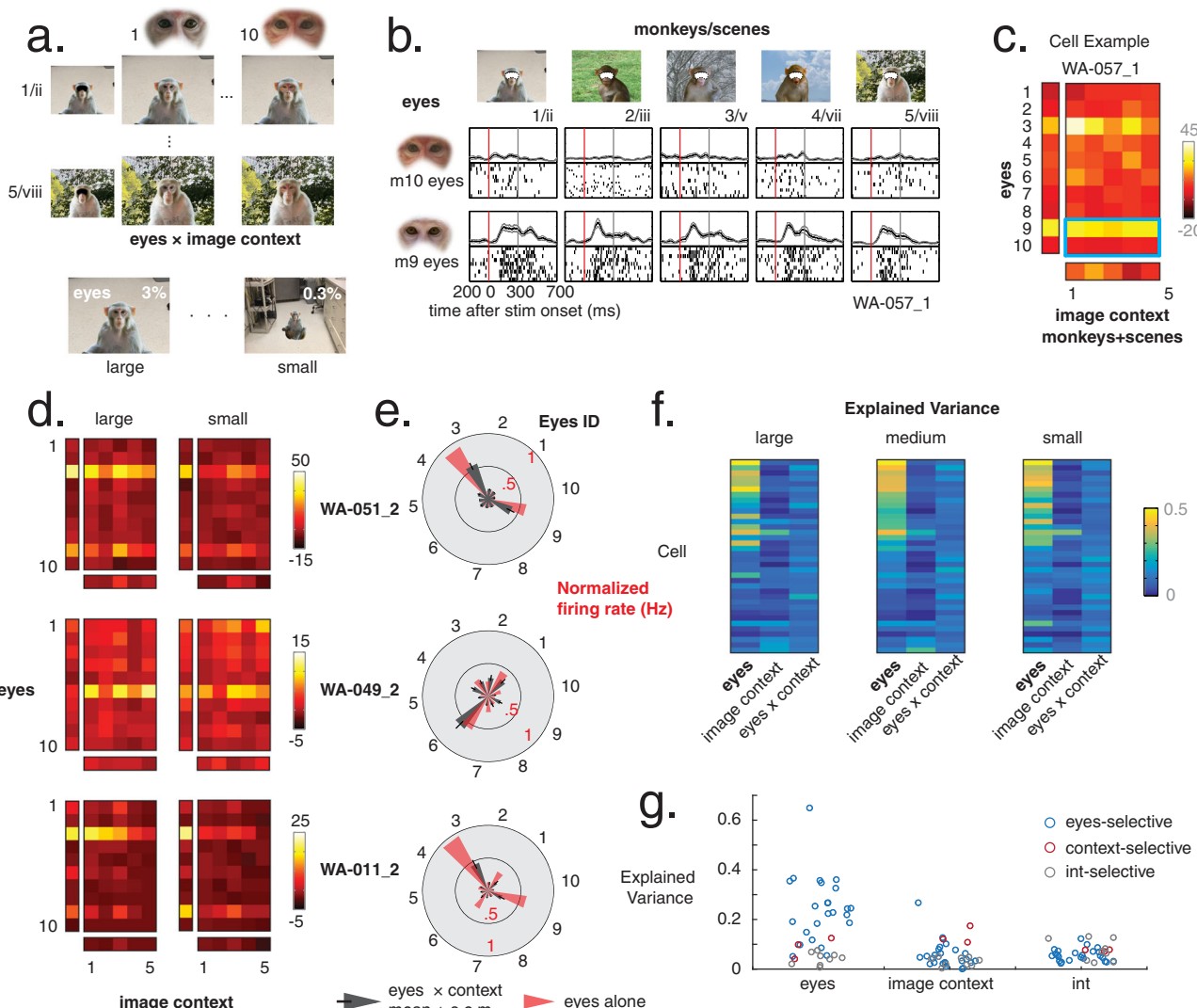

**Fig. 5 | Eye × Image Context swapping confirms dominance of local part tuning.**
**a** Recombination paradigm for eyes × image context stimuli. From full monkey ×
scene images, eyes from monkeys 1 to 10 are excised and exchanged with two full
image contexts. Images are presented at the same three sizes as in the original
experiment. **b** Raster plots for one eyes-selective AM cell, WA 057_1 (same as Fig. 2a,
b). Responses to eyes from monkeys 9 to 10 exchanged with five whole image
contexts. Eyes from m9 drive the response, with little input from the larger context.
*Macaque images in **a**–**b** were obtained from* https://figshare.com/articles/dataset/
Macaque_Faces/9862586/1 *under a CC-BY 4.0 license.* **c** Heatmap of average WA
057_1 responses to all large eyes × image context stimuli; tuned to eyes with low
contribution from image context (two-way ANOVA, explained variance due to

eyes = 0.36, image context = 0.03, int. = 0.05). **d** Heatmap responses for three more
AM neurons, each showing greater influence by eyes than image context or inter-
action. **e** Radial plots of eyes alone (mean, pink bar) and *eyes × whole image* (mean,
gray bar; SEM, black line) normalized responses for each identity, for cells in **d**.
Radial plots for all cells at this AM site, sorted by strength of eye tuning, shown in
Fig. S6. **f** Heatmaps of explained variance (two-way ANOVA) due to eyes, image
context, and interaction for stimuli of each size. Cells display strong tuning for eyes,
even as the size of the eyes within stimuli change up to tenfold. **g** Scatter plot of
explained variance for eyes, image context, and interaction; colored by tuning for
eyes (blue, $n = 24$), for context (red, $n = 3$), and interaction (gray, $n = 9$). The
responses of the majority of the cells were dominated by the eyes.

present findings, some data from human fMRI studies do not strongly
support the idea of increasing holistic processing in more anterior
nodes of the face processing network.

Studies assessing the consequences of damage to the ante-
rior temporal cortex are also informative. While such damage
affects some elements of face recognition, it does not seem to
strongly affect the perception of facial structure. For example,
human patients with damage to the anterior temporal cortex face
area often show associative forms of prosopagnosia in which they
cannot identify individuals, and perform poorly on tests of facial
memory[44]. However, many such patients largely retain their
capacity for the perceptual differentiation of individual faces[46],
suggesting that important aspects of structural analysis, includ-
ing those related to holistic processing, may be relatively

unaffected. Consistent with human neuropsychological cases,
lesions of macaque anterior inferotemporal cortex of the area TE,
where the AM face patch is located, caused only mild effects on
perceptual categorization of morphed faces[47]. Thus, while it may
be correct to say that anterior temporal face areas are closely
linked with individual recognition, the specific role of perceptual
and mnemonic mechanisms are still in question.

Psychophysical studies have emphasized the importance of hol-
istic processing in face perception[12,13]. If individual neurons in the
anterior face patches do not encode holistic features of faces, as our
findings suggest, how might the primate brain implement holistic
processing?

One possibility is that the holistic information is carried by the
anterior face patches but in a different manner. For example, a

previous study showed that the anterior inferotemporal cortex can encode the spatial configuration of facial parts by spike synchrony between individual neurons, not by spike firing rates[48]. Likewise, the neurons in the anterior face patches might encode the configural relationships of facial parts by spike synchrony among neurons which encode different facial parts.

A second possibility is that the face is processed holistically in different areas that are, in a sense, downstream from the face patches. One such candidate is the perirhinal cortex, which is located further anteromedial to the anterior face patches and heavily interconnected with visual areas in the inferotemporal cortices[49]. Perirhinal cortex was recently shown to be responsive to familiar faces[50], and the neurons in this region are capable of associating different visual components[51], which would lend the ability to combine multiple facial parts together to holistically encode known faces. A very recent study in macaques also identifies the temporal pole (TP) as responding strongly to familiar faces, closely matching psychophysical performance of face recognition[32]. Interestingly, in that study, face patch AM showed minimal correspondence with psychophysical performance, in some ways consistent with the present findings.

A third possibility is that the middle face patches, by virtue of their capacity to encode the context and spatial configuration of faces[8], play an important role in holistic face processing. Supporting this idea, facial inversion, which is thought to disrupt the holistic nature of face processing, significantly affects neuronal responses in the middle face patches but not in the anterior face patches[52]. Electrophysiological studies using a variant of the inverted face, the Thatcher-illusion, further showed that the majority of neurons in the middle face patches are susceptible to the illusion and were suggested to engage in the holistic processing of faces[53], while a relatively small population of neurons in the anterior inferotemporal cortices are sensitive to the illusion[54]. Human psychophysics has shown viewpoint-dependent primacy of holistic processing over parts-based processing[55,56] (but see ref. [57] for effects of viewpoint in part-based processing), which is consistent with the viewpoint-dependence of middle face patches[8].

Elements of each of these possibilities have been captured in recent theoretical models of face recognition[58-61]. For example, one computational model utilizes a hierarchical network that combines two sparse coding submodules for faces and objects to evaluate holistic and parts-based processing[59]. In this model, individual units became tuned to a small number of facial features, consistent with our results. Interestingly, holistic processing was then achieved through a top-down categorization that involved competition between the submodules. Such models can provide guidance on how observations such as ours might eventually be reconciled with more holistic elements of face recognition.

How does one reconcile the view of invariant identity coding in anterior face patch AM[9] with neural tuning dominated by local features? These findings are not necessarily in opposition. While we did not explore this facet, it is known that facial parts themselves carry significant information that can support viewpoint-invariant facial identification[62]. Thus, it is possible that the local tuning we observed can contribute to the view invariance observed previously.

Finally, as the local parts in our study themselves could theoretically be reduced further (e.g., eyes into pupil diameter, inter-eye distance, surrounding skin, etc.), it is possible that some form of conjunctive tuning or configural processing does contribute to the local tuning reported here, albeit restricted to configurations within relatively small subregions of the animal's face. For more conventional notions of holistic processing, however, our results demonstrate that individual anterior face patch neurons are more concerned with the encoding of specific subregions of the face and head, rather than their context within a face, head, body, or scene.

## Methods

### Subjects
Four rhesus macaques (WA (male, 8.5 kg), MA (male, 9.4 kg), SP (female, 8.5 kg), MO (male, 11 kg)) were used in this study. All monkeys were surgically implanted with an MRI-compatible head post, and with microwire electrode bundles (Microprobes) in face patches[29]. All surgeries were performed under aseptic conditions and general anesthesia under isoflurane, and animals were given postsurgical analgesics and prophylactic antibiotics. Monkeys WA and MA received 64 ch electrode implantations in left AM, monkey MO received a 64 ch electrode in left AF, and monkey SP received a 64 ch electrode in right AF. The electrode in monkey SP was used to record responses from an initial population of AF neurons, designated SP1, then lowered 125 μm to a new population of AF neurons, designated SP2, from which responses to the stimulus set were recorded again. All the procedures and animal welfare were in full compliance with the Guidelines for the Care and Use of Laboratory Animals by National Institute of Health and approved by the Animal Care and Use Committee of the National Institute of Mental Health.

### Targeting face patches
Functional Magnetic Resonance Imaging (fMRI) was performed on a 4.7 T vertical-bore MRI scanner (Bruker) with a gradient-echo echo planner imaging sequence. Hemodynamic responses were recorded, enhanced by the injection of MION. We located the AM and AF face patches with a standard fMRI block design, contrasting hemodynamic responses while the monkey viewed static images (monkeys MA, SP) or movies (monkeys WA, MO) of faces with responses while the monkey viewed objects, scenes and scrambled faces (Fig. 1f). fMRI data was analyzed with AFNI[63] and Matlab (Ver 2014b-2021b, MathWorks) software.

The stereotactic coordinates of each target patch were identified by overlaying the fMRI data with T1-weighted structural images which was taken with an MRI-compatible stereotaxic frame. With the guide of the stereotaxic system, we implanted a guide tube secured to a custom skull chamber and microdrive[29], through which we lowered a chronic microwire brush array. Before starting data collection, we lowered the electrode into the face patch over multiple days, and stopped when we found a position with a high proportion of channels holding face-selective cells. These microwire brush arrays allow neurons to be held for weeks on end[29], and this stability allowed us to hold neurons while collecting at least ten trials for each stimulus image (13,500 individual trials) over the course of 1–2 weeks.

### Electrophysiology recordings
Extracellular single-unit signals were recorded with the chronically implanted NiCr wires that permitted tracking of individual neurons over weeks[29,64]. The recorded neuronal signals were amplified and digitized at 24.4 kHz in a radio frequency-shielded room by PZ5 NeuroDigitizer (Tucker-Davis Technologies), and then stored to an RS4 Data Streamer controlled by an RZ2 BioAmp Processor (Tucker-Davis Technologies). A gold wire inserted into a skull screw was used for ground. Broadband signals (2.5–8 kHz) were collected from which individual spikes were extracted offline using the WaveClus software[65] after filtering between 300 and 5000 Hz. Event codes, eye positions, and a photodiode signal were also stored to a hard disk using OpenEX software (Ver 2.31.0) or Synapse software (Ver 92, Tucker-Davis Technologies).

The method for longitudinal identification of neurons across days was described in detail previously[29,64,66]. Briefly, spikes recorded from the same channel on different days routinely had

closely matching waveforms and inter-spike interval (ISI) histograms, and were provisionally inferred to arise from the same neurons across days. The trigger threshold for spike detection was manually set for each channel between 2.7 and 5.5 standard deviation of the filtered neuronal signal, the goal of satisfying two criteria: (1) maximizing the separation of spike clusters in principal component space and (2) consistent isolation quality, in terms of the cluster separation and ISI distribution, for the same channel across multiple days. The trigger threshold was adjusted if necessary to compensate for day-to-day changes in spike waveform and isolation.

This tentative classification based on waveform features and spike statistics was then tested against the pattern of stimulus selectivity and temporal structure of the neurons' firing evoked by visual stimulation using a set of 60 images of different categories (Fig. S2). Guided by our previous observations that neurons in inferotemporal cortex respond consistently to statically presented visual stimuli across days and even months[29,64], we used the distinctive visual response pattern generated by isolated spikes as a neural "fingerprint" to further disambiguate the identity of single units over time.

## Stimuli

We collected ten high-resolution photographs of macaque monkeys of various sexes and body positions, but all with heads facing directly toward the camera. *Due to copyright reasons, we display mock stimulus images in Figs.* 1, 2, 4, 5, *and* S1, *created from third-party images available under the Creative Commons open license.* T*he sources of these images are listed in Table* S2. *The original stimulus images used in the experiments can be provided upon request.* These images were resized such that the inter-eye distance was consistent across all images. We split these ten monkeys into four component parts: eyes, mouth, outer face, and body (Fig. 1a). The parts have a small overlap region with a gradation of transparency, which serves as a marginal zone for smooth recombination of images (see details below). We applied the same cropping window for eyes and mouth parts across images, while we performed image-specific extraction for the external margins of the outer face and body parts. To compensate for the variation of the border between the outer face and body, the hidden part of the body image around the neck was filled by content-aware filling tool of Photoshop software (Adobe software). Separately, we created a fifth image part, the background scene, and collected ten photographs of various indoor and outdoor scenes (representative scene stimuli shown in Fig. 1c). The scenes were chosen to be at a naturalistic distance when the monkey images were placed at the center of the scene, so that the monkeys were recognized in a normal size. We recombined these image parts into four "swapping" categories. All ten monkeys were combined with each of the ten background scenes, creating 100 unique *monkey × scene* images (Fig. 1d). Similarly, the *head × body*, *inner × outer face*, and *eye × mouth* swapping categories were created by excising each of the listed features from the original ten monkeys, and systematically exchanging them within each paired part to create 100 novel monkeys, heads, and inner faces, respectively (Fig. 1d). During the recombination, facial parts were blended with a half-transparent marginal zone to make a smooth transition. For the recombination of *head × body* and *monkey × scene* images, heads were placed over bodies and monkeys were placed over scenes. Each of the 450 images (400 recombined images + 50 image parts in isolation) was resized (Fig. 1e) to three different sizes, small (3° visual angle head width), medium (6°), and large (9°), creating a total of 1350 stimuli (representative stimuli shown in Fig. S1a).

In the follow-up *eyes × image context* experiment (Fig. 5), the eyes from the ten monkeys were recombined with five 'image contexts',

consisting of a full monkey except for the eyes (mouth, outer face, and body) against a background scene. These were resized to small, medium, and large sizes (as above), resulting in 210 stimuli (150 recombined stimuli, 60 parts alone stimuli). Stimulus images representative of those shown in the *eyes × whole image* experiment can be found in Supplementary Fig. 1b.

## Task

We trained four monkeys to passively view these swapping stimuli. Monkeys sat in a custom primate chair while head-fixed and viewing a 4 K OLED screen and fixating on a central white dot. Eye position was monitored with an infrared camera and Eyelink(Ver 2.31) software.

Stimuli were presented using NIMH MonkeyLogic (Ver 2.2.20, on Matlab 2018a) behind the fixation point, such that the dot always appeared directly between the eyes. Images from the full stimulus set were randomly interleaved and presented in a 400 ms ON, 300 ms OFF viewing paradigm, in 5 blocks of 4 stimuli. Trials were restarted if the monkey's fixation fell beyond 2° visual angle from of the central point (Fig. 1e). Animals received a juice reward for successful fixation at the end of each block. The entire dataset was considered complete when the monkey completed 13,500 trials.

In addition, we presented the same 60 "Fingerprinting" images (including human and monkey faces, monkey bodies, objects, and scenes) each day in order to ensure that single cell responses across days remained stable (Fig. S2).

## Cell selection

We identified the same neurons across multiple days by assessing the waveform similarity and the "Fingerprint" stimuli responses. From the 2 AF and 2 AM face patches, we recorded from 403 unique neurons (after cross-correlation spike timing analysis to eliminate duplicates). From this population, we identified 208 face-selective cells using two criteria. First, we performed a one-way ANOVA to test for a significant effect by stimulus on the baseline-subtracted neural response in at least one of 12 (4 groups × 3 sizes) swapping categories (Bonferroni corrected, $p < 0.0041$), to ensure that at least some of the specific stimuli would drive the neurons. For the second, a one-way ANOVA tested for a significantly different response (baseline-subtracted) for the heads and monkeys in our stimulus set, relative to the bodies and scenes in isolation from our image set.

The follow-up *eyes × image context* experiment was performed on monkey WA (left AM), and we collected recordings from 36 single units. No selection criteria were applied.

## Data analysis

The firing rate of each cell in response to a single stimulus was calculated by computing the average number of spikes between 70 and 350 ms after stimulus onset. Baseline firing rate was calculated as the average number of spikes between 150 ms before and 50 ms after stimulus onset. Firing rates were averaged across all trials of a single stimulus.

We performed a two-way analysis of variance (ANOVA) on the baseline-subtracted responses for all the stimuli within a swapping category (e.g., *mouth × eyes*), testing the variance explained by each part (e.g., eyes, mouth) and their interaction (Fig. 4). For our 'eyes-selective' population, we selected cells that had more variance explained by the response to eyes, inner face, head, and monkey, than by the mouth, outer face, head, or scene, respectively (and all relevant interaction terms).

We similarly separated 'mouth-selective' and 'outer-face-selective' populations. See Figs. 4c, S4a–f for further descriptions of these populations.

For the cell-sorting analysis (Fig. 4a–b), within the eyes-selective population we computed the average firing rate for the eyes across the recombined *eyes × mouth* stimuli, creating a 10 × 1 vector of *eyes*

responses. These vectors for all eyes-selective cells were ordered by finding the eyes identity that produced the highest firing rate, and sorting cells in order of the strongest identity preference (1–10) of each. The average *inner face, head*, and *monkey* responses were similarly computed for each cell, along with the average *mouth, outer face, body*, and *scene* responses. The cells in each of these other part-averaged response matrices were ordered according to the ordering by eyes preferences. Correlations were computed between the *eyes* response matrix and each *eyes*-sorted response matrix for the other parts. This was repeated for each local part across eyes-selective cells. The entire procedure was similarly performed for mouth-selective and outer-face-selective cells (Fig. S4g).

The regression analysis (Fig. 4d–f) for eyes-selective cells used the same average part responses as above (Fig. 4a), and compared the *eyes* responses (averaged across all mouth pairings) and the *mouth* responses (averaged across all eyes pairings) with the *monkey* responses (averaged across all scene pairings). A linear regression was performed between the *eyes* and *monkey* responses, and the *mouth* and *monkey* responses (independent from one another), and the regression coefficients were collected (Fig. 4f). Statistical difference between distributions of regression coefficients was computed using a Wilcoxon Signed Rank test. Regression coefficients for primary and secondary parts were similarly collected from mouth-selective and outer-face-selective cell populations.

We compared responses to features presented in isolation and in combination (Fig. S5) by computing, for each part-selective cell, (i) the response to the part alone (e.g., eyes alone) and to any features that contained that part (e.g., inner face alone, head alone, monkey alone), and (ii) the mean response to that part in combination (e.g., eyes across all eyes × mouth combinations, inner face across all inner × outer face combinations ….) across all sizes (Fig. S5a). For each stimulus, we plotted the difference in responses for the isolated and combined presentation, and plotted data from all cells across swapping groups, and cell types (Fig. S5b, top row). We additionally computed, for all stimuli for a single cell, each cell's mean difference between isolated and combined stimuli, and plotted a histogram of cell responses across swapping groups and cell types (Fig. S5b, bottom row).

The firing rate and ANOVA analyses for the *eyes × whole image* stimuli were calculated as above (as in Fig. 3). For the radial plots (Figs. 5e, S6), we calculated the firing rates for all ten eyes-alone stimuli, normalized to the highest and lowest stimulus responses for each cell. We also calculated the normalized firing rates for all ten eyes when recombined in *eyes × whole image* stimuli (averaged across all image context pairings. For each identity, we plotted the average eyes-alone response and average *eyes × whole image* response (averaged by eyes). Exemplars demonstrating response similarity are shown in Fig. 5e, and radial plots for all cells are shown in Supplementary Fig. 6. The latter are sorted in order of the magnitude of difference between variance explained by eyes and variance explained by image context (two-way ANOVA). Eyes-selective cells (higher response variance explained by eyes than by image context or interaction) labeled.

### Reporting summary
Further information on research design is available in the Nature Research Reporting Summary linked to this article.

## Data availability
The source data generated in this study have been deposited in a figshare database found at https://doi.org/10.6084/m9.figshare.19947188. Source data are provided with this paper.

## Code availability
The custom scripts used for data analysis in this study is available in a figshare repository at https://doi.org/10.6084/m9.figshare.19947182.

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

## Acknowledgements
Thanks to Katy Smith, David Yu, Frank Ye, Charles Zhu, Aidan Murphy, and Amit Khandhadia for assistance with this study. Functional and anatomical MRI scanning was carried out in the Neurophysiology Imaging Facility Core (NIMH, NINDS, NEI). This research was supported by the Intramural Research Program of the National Institute of Mental Health (ZIAMH002838 and ZIAMH002898). This work utilized the computational resources of the NIH High Performance Computing Core (https://hpc.nih.gov/).

## Author contributions
K.W.K. and D.A.L. designed experiment. K.W.K., J.J.H., and E.N.W. created stimuli. K.W.K., E.N.W., J.J.H., and B.E.R. trained monkeys, localized face patches, and implanted electrodes. K.W.K., E.N.W., and J.J.H. collected data. E.N.W. and K.W.K. performed data analysis. E.N.W., K.W.K., and D.A.L. prepared paper.

## Funding

## Competing interests
The authors declare no competing interests.
