## [Peer Review File · Nature Communications]

REVIEWER COMMENTS

Reviewer #1 (Remarks to the Author):

1. Summary

The authors address the question whether neurons in the anterior face patches (AM and AF) in macaque IT, previously thought to represent facial identity, are mainly driven by local facial features or holistic facial structure. Using a recent chronic, multi-unit recording technique (Microprobe), they collected response data to a large number of combinations of facial and body parts to reveal the contributions of each part to the overall responses. Their results indicate that the responses of the face-selective neurons in the anterior patches are predominantly explained by a single local facial feature rather than a combination of features.

2. Evaluation

The anterior patches have been thought to represent holistic combinations of facial features since previous physiological studies have shown identity selectivity in neurons therein and since higher vision is generally considered to increase view invariance and identity selectivity from posterior to anterior areas. In this sense, the results reported in the present study, opposing the previous view, are quite surprising. The experiments seem to have been conducted properly and the results look clear, convincing, and well presented.

I also see novelty in the method. The study relies on the swapping stimulus design, which allows for extensively exchanging facial and body parts in monkey images and thus requires a large number of stimuli to be presented (1350 images x 10 trials). For this purpose, the authors have introduced Microprobe (chronic, multi-unit recording) to collect clean response data. This is a nice demonstration of the new technology, which can potentially change the game in future research in this area.

In addition, their discussion section is quite informative and comprehensively citing relevant physiology studies. It also clarifies what the remaining questions are, which offers a clear vision for future investigation.

The manuscript is written with excellent clarity. I enjoyed very much reading through the entire manuscript.

Thus, I'm quite positive about this study. However, I'd like to request a minor revision in a few points that can improve the manuscript.

1) Explained variance plots (Figure 3c, d)

These plots are somewhat confusing since the comparison was made on the explained variances for eyes, mouth, and outer face, where the first two were calculated from the response data for "eyes x mouth" and the last one was from the data for "inner x outer faces". It seems a bit strange to show these explained variances in the same plot; at least, those values cannot directly be compared. The authors should reconsider whether using the 3D plot format is crucial. Perhaps, it might be better to show two separate 2D plots for "mouth x eyes" and "inner x outer faces". Or else, they should give a justification for the current format.

2) Discussion

The discussion section is generally well written and interesting, but I'd like to invite the authors to computational studies for the macaque face-processing system and discuss the relevance to the present study. In particular, the four theoretical studies listed below would be the most important. Among these, (c) should be the most relevant to the present study since it talks about parts-based and holistic processing in face-selective neurons. I would like to see discussion what the implication of these theories would be to the present experiment, and vice versa.

a. Leibo, J. Z., Liao, Q., Anselmi, F., Freiwald, W. A. & Poggio, T. View-tolerant face recognition and hebbian learning imply mirror-symmetric neural tuning to head orientation. *Curr. Biol.* 27,62–67 (2017).

b. Yildirim, I., Belledonne, M., Freiwald, W. & Tenenbaum, J. Efficient inverse graphics in biological face processing. *Sci. Adv.* 6, (2020).

c. Hosoya, H. & Hyvärinen, A. A mixture of sparse coding models explaining properties of face neurons related to holistic and parts-based processing. *PLoS Comp. Biol.* 13, e1005667 (2017).

d. Raman, R. & Hosoya, H. Convolutional neural networks explain tuning properties of anterior, but not middle, face-processing areas in macaque inferotemporal cortex. *Commun. Biol.* 3, 1–14 (2020).

3) Other minor comments

Using the alphabet "x" in the swapping categories like "eyes x mouth" looks a bit awkward, in particular, when it appears in italic font. I request that all "x" be replaced by a proper cross symbol.

Many figures look too small and sometimes details are hard to see. For example, in Figure 1e, fixation points and fixation windows are supposed to be drawn, but actually not visible. The authors should check carefully all the figures and resize them so that any important details should be viewable without stress (not on screen but in print).

It's not clear what the purpose of Figure 4d is from initial reading. At least, it's not specifically referred to from the text. I understand one-dimensional regression is performed between the corresponding elements of each vector. This point should be explicated.

Regarding the regression analysis (p. 16), I initially thought that the authors want to use a linear regression to predict monkey responses from eyes responses and I got confused since regressor estimation and testing are performed using the same entire data (10 data points). But I eventually understand that they simply want to predict monkey responses directly from eye responses, and measure the prediction performance by r-square and regression coefficients. A simple note on this point would help.

Figure 5e is rather confusing. What the radial and angular axes represent is described clearly enough. Figure S5 has helpful legends. These should be copied to Figure 5e.

In Methods, "steam pairing" should be "scene pairing"?

Reviewer #2 (Remarks to the Author):

In this manuscript, Waidmann et al tested whether the anterior face patches of the macaque temporal lobe, thought to be important for identity discrimination, primarily encode global or local facial features. They extracted the eyes, mouth, facial contour and body from natural images of 10 different individuals and systematically switched them, creating a large set of "recombinant" visual stimuli. The response

patterns of individual facial cells to these stimuli are best explained by selectivity for individual facial features, e.g., eyes or mouth, independent of the face or body in which they are embedded, and generally do not show selectivity for particular combinations of features, arguing against a holistic representation of facial identities in these cortical regions.

The study design, recording approach (fMRI-guided electrophysiology with chronically implanted probes), data quality, and analysis methods are excellent, and the results make an original and interesting contribution to the neural mechanisms of face processing.

Here are some remarks/questions/suggestions that the authors might consider.

1. Information on the proportion of cells coding for eyes, mouth, outer face in AM and AF in both areas would be welcome. Figure 3a gives us the impression that it is mainly the eyes in AM and the mouth in AF. Is this real or is it just a sampling bias? Also, how do you explain that the types of selectivity encountered in AF for monkeys MO and SP are so different?

2. Outer face cells account for almost half of the selective neurons (Fig 3d) but are somewhat overlooked. The results of the correlation analysis, which illustrate how well responses to individual features predict neuronal activity across the different image contexts, are not as impressive for the outer-face-selective cells as they are for eyes- and mouth-selective cells. What could be the reason for this?

3. Figure S5 shows many examples of cells with very similar eye tuning patterns for isolated and integrated eyes, but there are also many examples where the match is weak or completely absent. Some cells also appear to show sharper tuning for integrated eyes than for isolated eyes. How should such patterns be interpreted?

4. The eyes and mouth appear to be considered basic facial features, but this may be somewhat exaggerated. The eye region combines the shape and size of the eyes, the shape of the eyebrows, the interocular distance, and information about skin color. The mouth region includes a specific combination of lip, nose and chin shapes. These body parts are rich in information, can be considered configurations in their own right and could play an important role in defining facial identity.

5. Curiously, given the results of this study, much of the discussion is an exercise in saving the holistic soldier. There is ample psychophysical evidence that configural information plays a role and thus must be represented in some way in the cortical face-processing system, but the authors may also want to reevaluate the relative importance of holistic versus feature information in face identity recognition. If

empirical data are lacking, psychophysical experiments using the type of stimuli developed for the present study might help answer this question.

6. The legend for Figure S1 is misleading and incomplete. There are not 1350 stimuli on S1.a but 108, and I believe the row and column references are reversed. Also, there is no legend for S1.c and S1.d.

7. Figure S2.c. Please clarify the meaning of the colored waveforms. Is each trace the average waveform over a given session? From Figure S2.d, 5 sessions are represented, so some waveforms must be perfectly overlapped.

REVIEWER COMMENTS

Reviewer #1 (Remarks to the Author):

1. Summary

The authors address the question whether neurons in the anterior face patches (AM and AF) in macaque IT, previously thought to represent facial identity, are mainly driven by local facial features or holistic facial structure. Using a recent chronic, multi-unit recording technique (Microprobe), they collected response data to a large number of combinations of facial and body parts to reveal the contributions of each part to the overall responses. Their results indicate that the responses of the face-selective neurons in the anterior patches are predominantly explained by a single local facial feature rather than a combination of features.

2. Evaluation

The anterior patches have been thought to represent holistic combinations of facial features since previous physiological studies have shown identity selectivity in neurons therein and since higher vision is generally considered to increase view invariance and identity selectivity from posterior to anterior areas. In this sense, the results reported in the present study, opposing the previous view, are quite surprising. The experiments seem to have been conducted properly and the results look clear, convincing, and well presented.

I also see novelty in the method. The study relies on the swapping stimulus design, which allows for extensively exchanging facial and body parts in monkey images and thus requires a large number of stimuli to be presented (1350 images x 10 trials). For this purpose, the authors have introduced Microprobe (chronic, multi-unit recording) to collect clean response data. This is a nice demonstration of the new technology, which can potentially change the game in future research in this area.

In addition, their discussion section is quite informative and comprehensively citing relevant physiology studies. It also clarifies what the remaining questions are, which offers a clear vision for future investigation.

The manuscript is written with excellent clarity. I enjoyed very much reading through the entire manuscript.

We thank this reviewer for this positive evaluation of our study.

Thus, I'm quite positive about this study. However, I'd like to request a minor revision in a few points that can improve the manuscript.

1) Explained variance plots (Figure 3c, d)

These plots are somewhat confusing since the comparison was made on the explained variances for eyes, mouth, and outer face, where the first two were calculated from the response data for "eyes x mouth" and the last one was from the data for "inner x outer faces". It seems a bit strange to show these explained variances in the same plot; at least, those values cannot directly be compared. The authors should reconsider whether using the 3D plot format is crucial. Perhaps, it might be better to

show two separate 2D plots for "mouth x eyes" and "inner x outer faces". Or else, they should give a justification for the current format.

We appreciate the reviewer's suggestion and have replaced the 3-dimensional plots describing the eye-selective, mouth-selective, and outer face-selective populations with 2-dimensional plots of the response variance explained by eyes and mouth for eyes- and mouth- selective cells, and inner face and outer face for outer face selective cells (previously Fig. 3d, now Fig. 3c). We expand on this in the Results section on page 11:

Upon separating subpopulations that showed any explained variance preference for each of these parts (135/208 of the face selective neurons, see Methods), cells demonstrated highly selective tuning to these single local features (Fig. 3c). The eyes- and mouth-selective cells both displayed minimal tuning for the opposite internal facial feature, and the outer face-selective cells showed low tuning to the internal face.

We did retain the single full-population 3D scatter plot, and moved it to the end of the figure (previously Fig. 3c, now Fig. 3d). The two explained variance calculations (mouth x eyes, inner x outer faces) were made on the same group of cells, performed the same way for each calculation, and plotted on equivalent scales. Therefore we feel the same plot can be used to *visually* depict the behavior of the population as a whole. We appreciate the reviewer's point here, and we now clarify in the text that this plot is for the purposes of visualization (page 11 of the Results):

The variance explained by eyes, mouth, and outer face for every face-selective cell was further plotted for visualization purposes in Fig. 3d, and the responses of most cells followed one of the three axes, showing little conjoint selectivity.

2) Discussion

The discussion section is generally well written and interesting, but I'd like to invite the authors to computational studies for the macaque face-processing system and discuss the relevance to the present study. In particular, the four theoretical studies listed below would be the most important. Among these, (c) should be the most relevant to the present study since it talks about parts-based and holistic processing in face-selective neurons. I would like to see discussion what the implication of these theories would be to the present experiment, and vice versa.

a. Leibo, J. Z., Liao, Q., Anselmi, F., Freiwald, W. A. & Poggio, T. View-tolerant face recognition and hebbian learning imply mirror-symmetric neural tuning to head orientation. *Curr. Biol.* 27,62–67 (2017).

b. Yildirim, I., Belledonne, M., Freiwald, W. & Tenenbaum, J. Efficient inverse graphics in biological face processing. *Sci. Adv.* 6, (2020).

c. Hosoya, H. & Hyvärinen, A. A mixture of sparse coding models explaining properties of face neurons related to holistic and parts-based processing. *PLoS Comp. Biol.* 13, e1005667 (2017).

d. Raman, R. & Hosoya, H. Convolutional neural networks explain tuning properties of anterior, but not middle, face-processing areas in macaque inferotemporal cortex. *Commun. Biol.* 3, 1–14 (2020).

We thank the reviewer for these suggestions and have now added a paragraph to the discussion (p. 21) drawing attention to how theoretical models of face recognition might help understand the present results, including the references indicated. (58-61).

Elements of each of these possibilities have been captured in recent theoretical models of face recognition⁵⁸⁻⁶¹. For example, one computational model utilizes a hierarchical network that combines two sparse coding submodules for faces and objects to evaluate holistic and parts-based processing⁵⁹.

In this model, individual units became tuned to a small number of facial features, consistent with our results. Interestingly, holistic processing was then achieved through a top-down categorization that involved competition between the submodules. Such models can provide guidance on how observations such as ours might eventually be reconciled with more holistic elements of face recognition.

3) Other minor comments

Using the alphabet "x" in the swapping categories like "eyes x mouth" looks a bit awkward, in particular, when it appears in italic font. I request that all "x" be replaced by a proper cross symbol.

We have replaced any "x" text with the cross symbol " × " in all text and figures.

Many figures look too small and sometimes details are hard to see. For example, in Figure 1e, fixation points and fixation windows are supposed to be drawn, but actually not visible. The authors should check carefully all the figures and resize them so that any important details should be viewable without stress (not on screen but in print).

All figures have been resized, and some figure text darkened or enlarged, to make their reading on print easier.

It's not clear what the purpose of Figure 4d is from initial reading. At least, it's not specifically referred to from the text. I understand one-dimensional regression is performed between the corresponding elements of each vector. This point should be explicated.

Regarding the regression analysis (p. 16), I initially thought that the authors want to use a linear regression to predict monkey responses from eyes responses and I got confused since regressor estimation and testing are performed using the same entire data (10 data points). But I eventually understand that they simply want to predict monkey responses directly from eye responses, and measure the prediction performance by r-square and regression coefficients. A simple note on this point would help.

We hope the addition of the following text to the Results section (page 14-15) and slight modifications to Fig. 4d help clarify these analyses.

In the eye- selective example neuron shown in Fig. 4d-e, the average responses to each pair of eyes closely resembled the responses to the full monkeys containing those eyes ($r^2 = 0.92$). By contrast, for the same group of eye-selective neurons, the average responses to each mouth were much less predictive of the corresponding responses to the full monkeys, though still positively correlated ($r^2 = 0.55$). For each cell, we performed a linear regression comparing the responses to full monkeys to the average responses to primary features (eyes, mouth, outer face) for each tuning type (eyes-selective, etc.), to ask how well a response to one set of eyes embedded in just an inner face, for example, would predict the response to a full monkey image with those eyes. Across the population of eyes-, mouth-, and outer face-selective neurons, the average regression coefficients comparing the primary features against monkey responses were near 1.0, indicating that the responses of neurons to full monkey identity were not only qualitatively predicted by local features but also quantitatively predicted (Fig. 4f).

Figure 5e is rather confusing. What the radial and angular axes represent is described clearly enough. Figure S5 has helpful legends. These should be copied to Figure 5e.

We have changed the Fig. 5e legend accordingly.

In Methods, "steam pairing" should be "scene pairing"?

This typo has been corrected.

Reviewer #2 (Remarks to the Author):

In this manuscript, Waidmann et al tested whether the anterior face patches of the macaque temporal lobe, thought to be important for identity discrimination, primarily encode global or local facial features. They extracted the eyes, mouth, facial contour and body from natural images of 10 different individuals and systematically switched them, creating a large set of "recombinant" visual stimuli. The response patterns of individual facial cells to these stimuli are best explained by selectivity for individual facial features, e.g., eyes or mouth, independent of the face or body in which they are embedded, and generally do not show selectivity for particular combinations of features, arguing against a holistic representation of facial identities in these cortical regions.

The study design, recording approach (fMRI-guided electrophysiology with chronically implanted probes), data quality, and analysis methods are excellent, and the results make an original and interesting contribution to the neural mechanisms of face processing.

We thank this reviewer for this very positive comment about our study.

Here are some remarks/questions/suggestions that the authors might consider.

1. Information on the proportion of cells coding for eyes, mouth, outer face in AM and AF in both areas would be welcome. Figure 3a gives us the impression that it is mainly the eyes in AM and the mouth in AF. Is this real or is it just a sampling bias? Also, how do you explain that the types of selectivity encountered in AF for monkeys MO and SP are so different?

We have added new text in the Results section (page 11-12) to describe the populations across patches: **These three populations (see Fig. S4a-b for the parts-selective and non-parts-selective units) were not strictly divided between the two face patches. While eyes-selective units predominated at both AM recording sites (22/44 MA, 24/36 WA), outer face- and mouth-selective cells were also present in smaller proportions (Table S1). Face patch AF was more heterogeneous in its neural tuning profiles and differed somewhat across AF recording locations, with mouth-selective cells predominating at one site (19/36 MO) and outer face cells at the others (19/37 SP1, 25/55 SP2) and other selectivity types minimally represented (Table S1). However, across all sites the single local features dominated the responses to faces across dramatic changes in image context, with little evidence for combinatorial or holistic tuning.**

and the Discussion section (page 18) to discuss the particular heterogeneity of the AF sites:

The most common tuning profile in patch AM involved the upper face, with more varied tuning profiles within AF (Fig. 3a, Table S1). Heterogeneous selectivity for different facial parts was also previously reported in ML⁸ and AM³² face patches. The AF tuning was more heterogeneous and also varied somewhat across recording sites, which may reflect different functional subdivisions within the

face patch, or differences in the specialization across animals. ... In sum, the AM and AF showed potentially important differences in the specific features to which they were tuned, but crucially, neither appeared to be an integration site for holistic face information.

We also added a table (Table S1, page 41) containing the percentages of each tuning type for every monkey. At each site there does appear to be one feature that many more of the cells are tuned to. We believe this is not solely due to sampling bias, because to find local part selective cells, we selected for cells that had any preference (no matter how large or small) for monkeys, heads, inner or outer face, and eyes or mouth. Each site contains cells tuned for at least 2 different features (though in different proportions) and we included an additional data panel (Figure S4a) to display the variance explained by all swapping factors for parts-selective cells included (135/208) and non-selective cells excluded (73/208) in our analysis. The majority of excluded cells appear to show no selectivity for any of the swapping features in our stimulus set, except for a small population that showed some tuning to outer face, but not to heads or monkeys (and still minimal tuning to inner face or the interaction), so the preferences of that group of cells is unclear.

2. Outer face cells account for almost half of the selective neurons (Fig 3d) but are somewhat overlooked. The results of the correlation analysis, which illustrate how well responses to individual features predict neuronal activity across the different image contexts, are not as impressive for the outer-face-selective cells as they are for eyes- and mouth-selective cells. What could be the reason for this?

We welcome the reviewer's assessment of the outer face patch tuning and now place more emphasis on this. The perception of a weaker effect for the outer face in the correlation matrix (Formerly Fig. S4f, now S4g) was, in part, due to the ordering of the correlation matrix. We rearranged the tiles, and also changed the color axis (formerly [-1, 1]) to [-0.5, 1] to match the one used for the eyes-selective cells correlation matrix in the main figure (Fig. 4c). While the correlation values are slightly weaker for the outer face-selective cells than for the eyes- and mouth-selective populations, there is still the strongest correspondence between the outer face response and the head and monkey responses. We also briefly discuss the outer face further in new Discussion text below (page 18)

The high proportion of the outer face-selective neurons in AF was unexpected, in part because the fundus of the STS is thought to be specifically adapted for configurable facial features that confer dynamic social information, most notably the eyes and mouth³³⁻³⁶. The outer face represents a fixed and immutable cue for face identity, and, aside from the ears and some postural cues, has only a secondary role in social signaling³⁷. While the specificity of outer face tuning was slightly weaker than for eyes and mouth (see Fig. 3c), the responses of many AF neurons were dominated by outer face information in the recombination and transplantation paradigms. Since the external features of faces are more important for face processing in young children^{38,39} and configural processing for the internal face develops slower than feature-based face processing⁴⁰, the neurons in AM and AF face patches may reflect a perceptual mechanism which develops earlier in the brain. Future investigations should consider the outer face as a significant contributor to neural tuning in this region.

3. Figure S5 shows many examples of cells with very similar eye tuning patterns for isolated and integrated eyes, but there are also many examples where the match is weak or completely absent. We did not perform any cell-selection process for the population in the original Figure 5 and S5, so these figures contain the entire population regardless of tuning type. We recorded 36 units during this session from monkey WA, and of those, 24 (66%) show tuning for eyes (higher explained variance for eyes than for image context and interaction; similar to previous tuning definitions). An additional panel (Fig. 5g) has been added to show the explained variance for the populations tuned more for eyes, image context,

or their conjunction. As in the original stimulus paradigm, there is a range of tuning strengths present as we do not threshold by strength of tuning or degree of explained variance preference for eyes over other features. Importantly, tuning to the image context or interaction is still consistently lower across this population.

We also changed Fig. S6 (formerly Fig. S5) to label cells that were categorized as tuned to eyes, image context, or interaction, and adjusted the normalization so the scales for eyes alone and eyes combined were more directly comparable. As noted, while there are many strong relationships between eyes alone and combined responses, there are others in the eyes-selective population that show a weak or apparently absent relationship. This could be due to either a very weak level of eyes-selectivity, or a difference in encoding of isolated vs. combined stimuli.

Some cells also appear to show sharper tuning for integrated eyes than for isolated eyes. How should such patterns be interpreted?

We thank the reviewer for this observation, and have included an additional supplementary analysis (Fig. S5, Fig. S6b) to examine any relationship between the isolated and combined features.

For responses to the main experimental stimulus set, for every cell across all three of the part-selective types (eyes-selective, mouth-selective, and outer face-selective cells) we computed (i) the mean normalized firing rate for every part alone (eyes-selective cells: eyes alone, inner face alone, head alone, and monkey alone) and (ii) the mean normalized firing rate across stimuli in which those features are presented in combination (i.e. eyes-selective cells: average eyes across eyes x mouth, average inner face across inner face x outer face, ...).

When we plot the difference between responses to isolated and combined stimuli (Fig. S5b), we can observe that for each tuning type across swapping groups, the majority of isolated and combined stimuli are represented roughly similarly, with a range of preferences between parts presented in isolation or combination. The distributions are mostly situated around 0, indicating that most cells show no preference between parts alone and parts combined. There are slight but statistically significant (t-test, $p < 0.01$) population-level (Fig. S5b, top row) and cell-level (Fig. S5b, bottom row) tendencies to prefer combined stimuli among the mouth- and outer-face selective cells for some swapping categories, possibly indicating that these face-selective neurons prefer full faces, while the identity tuning remains driven by the local features. We have added the following text explaining this result on page 16:

We also directly compared responses to isolated and combined parts stimuli for all parts-selective cells (Fig S5a). There was a range of preference for isolated or combined stimuli, with most of the parts stimuli represented roughly similarly across both conditions (Fig. S5b). Eyes-selective cells as a whole showed little preference between isolated and combined stimuli. Mouth-selective and outer face-selective cells showed a slight but significant (t-test, $p < 0.01$) preference for features in combination for some swapping groups (Fig S5b), while still owing their identity tuning primarily to the mouth or outer face.

4. The eyes and mouth appear to be considered basic facial features, but this may be somewhat exaggerated. The eye region combines the shape and size of the eyes, the shape of the eyebrows, the interocular distance, and information about skin color. The mouth region includes a specific combination of lip, nose and chin shapes. These body parts are rich in information, can be considered configurations in their own right and could play an important role in defining facial identity.

We thank the reviewer for this comment, and we agree. We have added the following text in the Introduction (page 6) to clarify this point so that readers do not draw the wrong conclusion:

The term ‘eyes’ was used to label the upper region of the inner face, including the eyes, portions of the surrounding skin, eyebrows, and forehead. The term ‘mouth’ pertained to the lower inner face,

including the lips, jaw, and nose (Fig. 1a). While it is possible to dissect these regions further, the current study used upper and lower inner face components in its combinatorial parts analysis.

....and in the Discussion (page 20-21) :

Finally, as the local parts in our study themselves could theoretically be reduced further (e.g. eyes into pupil diameter, inter-eye distance, surrounding skin, etc.), it is possible that some form of conjunctive tuning or configural processing does contribute to the local tuning reported here, albeit restricted to configurations within relatively small subregions of the animal's face. For more conventional notions of holistic processing, however, our results demonstrate that individual anterior face patch neurons are more concerned with the encoding of specific subregions of the face and head, rather than their context within a face, head, body, or scene.

5. Curiously, given the results of this study, much of the discussion is an exercise in saving the holistic soldier. There is ample psychophysical evidence that configural information plays a role and thus must be represented in some way in the cortical face-processing system, but the authors may also want to reevaluate the relative importance of holistic versus feature information in face identity recognition. We have restructured the discussion somewhat, not to hang onto old views but to reconcile seemingly disparate pieces of evidence. We have also reworded the introduction a bit to match this emphasis. We note only that the large amount of psychophysical evidence for holistic processing suggests that it must be encoded *somewhere* in the primate brain, but that our specific question was whether AM and AF were performing holistic processing, as had been suggested, tacitly and explicitly, by the existing literature. As we find more evidence for parts-based processing in these specific regions, we contend that the seat(s) of holistic integration of face information may reside elsewhere.

If empirical data are lacking, psychophysical experiments using the type of stimuli developed for the present study might help answer this question.

We agree that our results suggest that a combination of psychophysics and electrophysiological recordings of neural activity would provide powerful insight into how the specific encoding of facial parts and whole faces by individual regions and neurons relates to the actual perception of faces. We will make our stimuli and stimulus creation method available for other researchers upon request, although for us to implement these specific experiments from scratch would be too intensive, with designing a novel psychophysics paradigm and training multiple animals on it, then implanting electrodes and collecting simultaneous electrophysiological data. We also feel that linking our results with psychophysics is beyond the scope of our study, as we set out to answer whether AM and AF were the seat of holistic processing, and not to link their activity with the larger percept itself.

6. The legend for Figure S1 is misleading and incomplete. There are not 1350 stimuli on S1.a but 108, and I believe the row and column references are reversed. Also, there is no legend for S1.c and S1.d. We apologize for this error; a previous version of this figure presented all 1350 stimuli on one page, but they were impossible to see individually in PDF form, and we replaced them with more visible exemplars. We have updated the legend for Fig. S1 to reflect the current figure layout.

7. Figure S2.c. Please clarify the meaning of the colored waveforms. Is each trace the average waveform over a given session? From Figure S2.d, 5 sessions are represented, so some waveforms must be perfectly overlapped.

This is correct, each waveform represents a different session, and there are some very closely overlapped session waveforms (visible by zooming in very closely). We have updated Fig S2c and its legend to clarify this.

REVIEWERS' COMMENTS

Reviewer #1 (Remarks to the Author):

All my previous comments are properly dealt with. I consider the current manuscript ready for publication.

The figures from page 46 are out of bounds. I suppose this will be resolved by the copy-editors.

Reviewer #2 (Remarks to the Author):

I am satisfied with this revised version. The authors have done an thorough revision. I recommend the publication of this work.

REVIEWERS' COMMENTS

Reviewer #1 (Remarks to the Author):

All my previous comments are properly dealt with. I consider the current manuscript ready for publication.

The figures from page 46 are out of bounds. I suppose this will be resolved by the copy-editors.

Reviewer #2 (Remarks to the Author):

I am satisfied with this revised version. The authors have done an thorough revision. I recommend the publication of this work.

We thank both reviewers for their comments and helpful suggestions.